# A Decision Support System for Face Sketch Synthesis Using Deep Learning and Artificial Intelligence

**DOI:** 10.3390/s21248178

**Published:** 2021-12-08

**Authors:** Irfan Azhar, Muhammad Sharif, Mudassar Raza, Muhammad Attique Khan, Hwan-Seung Yong

**Affiliations:** 1Department of Computer Science, COMSATS University Islamabad, Wah Campus, Wah Cantt 47040, Pakistan; irfanazhar681@gmail.com (I.A.); mudassarr@gmail.com (M.R.); 2Department of Computer Science, HITEC University Taxila, Taxila 47080, Pakistan; attique.khan@hitecuni.edu.pk; 3Department of Computer Science & Engineering, Ewha Womans University, Seoul 03760, Korea; hsyong@ewha.ac.kr

**Keywords:** smart cities, sketch synthesis, convolutional neural network, Vgg-19 net, U-Net, Spiral-Net, face recognition, NLDA, OpenBR

## Abstract

The recent development in the area of IoT technologies is likely to be implemented extensively in the next decade. There is a great increase in the crime rate, and the handling officers are responsible for dealing with a broad range of cyber and Internet issues during investigation. IoT technologies are helpful in the identification of suspects, and few technologies are available that use IoT and deep learning together for face sketch synthesis. Convolutional neural networks (CNNs) and other constructs of deep learning have become major tools in recent approaches. A new-found architecture of the neural network is anticipated in this work. It is called Spiral-Net, which is a modified version of U-Net fto perform face sketch synthesis (the phase is known as the compiler network C here). Spiral-Net performs in combination with a pre-trained Vgg-19 network called the feature extractor F. It first identifies the top *n* matches from viewed sketches to a given photo. F is again used to formulate a feature map based on the cosine distance of a candidate sketch formed by C from the top *n* matches. A customized CNN configuration (called the discriminator D) then computes loss functions based on differences between the candidate sketch and the feature. Values of these loss functions alternately update C and F. The ensemble of these nets is trained and tested on selected datasets, including CUFS, CUFSF, and a part of the IIT photo–sketch dataset. Results of this modified U-Net are acquired by the legacy NLDA (1998) scheme of face recognition and its newer version, OpenBR (2013), which demonstrate an improvement of 5% compared with the current state of the art in its relevant domain.

## 1. Introduction

The Internet of Things (IoT) [1] has been playing a key role in the smart city sector, for example, in the security of smart homes, where you, using your smartphone, can decide who can enter your home [2]. Through IoT technology, it is easy to monitor your home at any time from anywhere, and this process helps to develop efficient, safer smart cities [3]. The IoT technology integration with IT devices helps to ease the investigation process, especially in the identification of people [4,5]. A very few studies are available on how IoT and information technology (IT) techniques work together [6]. The major applications where these technologies work together are biometric [7], video surveillance [8], Internet of Vehicles [9], and biomedical [10,11].

The formulation of face sketches based on learning from the reference photos and their corresponding forensic sketches has been an active field since the last two decades [12,13]. It helps the law enforcement agencies in the search, isolation, and identification of suspects by enabling them to match sketches against possible candidates from the mug-shot library [14,15,16] and/or photo dataset of the target population [17,18]. Forensic or artist sketches are also used in animated movies and/or during the development of CGI-based segments [19]. Presently, many persons like to use a sketch in place of a personal picture as an avatar or a profile image. Therefore, a ready-made scheme to furnish a sketch from a personal picture, without involving a skilled sketch artist, would come in handy [20]. Since 2004, exemplar-based techniques incorporating patch-matching algorithms have been most popular. Photos and corresponding sketches were identically divided into a mosaic of overlapping patches. For each patch of the photo, its nearest patch in all training sketches according to a given property, for example, the Markov random field (MRF) [21], the Markov weight field (MWF), or spatial sketch denoising (SSD), was searched for and marked. This principle was applied successively to all photos and sketches in the training set. Hence, a dictionary was developed. For each test photo patch, a suitable patch was first searched for in the photo and its corresponding patch in the dictionary was selected as part of the resulting sketch [22]. On completion of this search, a resulting sketch was formulated. In previous research, much effort has been devoted to reducing the time spent on and resource overheads of these methods to effectively produce a sketch. Those algorithms did not focus on capturing the subtle non-linearity between the original photo and the forensic sketch. Their results were, however, only reliable for a dataset of subjects devoid of the diversity of ethnicity; age; facial hair; and external elements, such as earrings, glasses, and hairpins. While those methods could replicate major features of the test photo, they did not reproduce minor details, such as contours of the cheekbones, edges of mustaches/beards/hairstyles, or clear outlines of eyeglasses. Lately, neural networks and other tools of deep learning have been employed to learn about the correspondence between photo–sketch pairs, and they try to reproduce intricate features of the photo in the resulting sketch. These methods also have their small inadequacies. Simple CNN-based methods produce sketches that lack sharpness and focus [23,24]. On the contrary, GAN-based methods do produce clear sketches but they are incomplete concerning the outline of the test subject’s photo. This paper includes the following:
A novel/modified structure of a residual network with skip connections forming a spiral-like shape to act as a compiler entity in the proposed face sketch synthesis phase. The overall scheme is motivated by [25], and a similar approach is presented in [26].A pre-trained Vgg-19 network is used to help accomplish the exemplar-based technique of selecting the best possible candidate from the viewed sketches during the training process. This part relies upon the distribution of the input photo into a mosaic of overlapping patches and identical division of the sketches in the reference set.The patches are selected by the minimal cosine distance, and a candidate feature map of the sketch is formulated.The feature sketch and the raw sketch by the compiler network are then compared through a customized convolutional neural network applying the MSE loss function to render a perceptual loss that monitors the training of the compiler network.The adversary loss function is also used to give sharpness to the resulting sketches.

The rest of the paper is arranged in this sequence: Section 2 covers the previous and current works related to the proposed model. Section 3 describes the composition detail of the suggested network. Section 4 provides implementation details and discusses the evaluation and analysis of results. Section 5 gives the conclusion.

## 2. Related Work

The Internet of Things (IoT) and machine learning have shown improved performance in many applications, such as facial recognition, biometrics, and surveillance [27,28]. Recently, the blockchain-based multi-IoT method was presented by Jeong et al. [29]. The presented method works in two layers (layer and layer) with the help of the blockchain technology. Through these layers, information is sent to and received from local IoT groups in more secure ways. Another multi-IoT method was presented by [30] for anomaly detection. They introduced forward and inverse problems to investigate the dependency of the inter-node distance and the size of the IoT network. A new paradigm, named social IoT, was presented by Luigi et al. [31] for the identification of useful guidelines for institution and social management. Khammas et al. [32] presented a cognitive IoT approach to human activity diagnosis. In cognitive computing, the cognitive IoT is the next step to improving the accuracy and reliability of the system. An IoT-based biometric security system was presented by Bobby et al. [11]. In this system, the IoT allows the multiple sensors and scanners to interact with human beings.

The recent developments in the CNN for scene recognition [33], object recognition [34], and action recognition [35] have produced an impressive performance [36]. Tang and Wang [37] introduced in their seminal work a new art of formulating human face sketches based on Eigen transformation. The work is based on pairs of photos and their corresponding viewed sketches. They developed a correlation between input photos and training photos in the Eigenspace. Then, using this correlation, they proposed to construct a sketch from the Eigenspace of the training sketches. Liu et al. [38] proposed the non-linear model of sketch formulation based on locally linear embedding (LLE). In this model, the input photo is divided into overlapping patches. Then, each patch is reshaped by a linear combination of training patches. The same relationship of photo patches was used to formulate respective patches of the resulting sketch. Tang and Wang [39] used Markov random fields (MRF) in the selection of neighboring patches and to improve their relationship. Zhou et al. [40] proposed a model of sketch generation that further builds upon the MRF model. They added weights to linear combinations of best possible candidate patches, and it was called the Markov weight field (MWF). Song et al. [17] presented a model based on spatial sketch denoising (SSD). Gao et al. [41] proposed an adaptive scheme based on the practical benefits of sparse representation theory, and it was called the SNS-SRE method, which relates to sparse neighbor selection and sparse-representation-based enhancement. Wang et al. [42] formulated a solution of neighbor selection by building up a dictionary based on a random sampling of the training photos and sketches. This model was called random sampling and locality constraint (RSLCR). Akram et al. [43] carried out a comparative study of all basic methodologies of the exemplar-based approach as well as two newer methods of sketch synthesis, called FCN [44] and GAN [45], which are based on the convolutional neural network and generative adversarial networks, respectively. The last two works may be included among the pioneer efforts of “learning-based” algorithms of sketch synthesis. Zhang et al. [46] introduced a model to address the problems of texture loss of the FCN setup. Their scheme consisted of two-branched FCN. One computed a content image, and the second branch calculated the texture of the synthesized sketch. This model also inherited the inadequacy of distorted sketches since the two-branched network could not present a well-unified output. Wang et al. [47] proposed a model to generate sketches from training photos and photos from the training sketches by employing a multiscale generative adversarial network. Wang et al. [48] proposed a model of anchored neighborhood index (ANI) that incorporated correlation of photo patches as well as sketch patches during sketch formulation. Moreover, similar to RSLCR, this algorithm also benefited from the development of an off-line dictionary to reduce computational overheads during the testing phase. Jiao et al. [49] presented a deep learning method based on a small CNN and a multilayer perceptron. This work was successful in imparting continuous and faithful facial contours of the input photo to its resulting sketch. Zhang et al. [50] proposed a model based on adversarial neural networks that learned in photo and sketch domains with help of intermediate entities called latent variables. Synthesized sketches of this model bear improvement against blurs and shape deformations. Zhang et al. [51] proposed a model called dual transfer face sketch-photo synthesis (FSPS). It is based on CNN and GAN and realizes inter-domain and intra-domain information transfer to formulate a sketch from the training pairs of photo-viewed sketches. Lin et al. [52] and Fang et al. [53] presented individual works based on neural networks for face-sketch formulation involving the identity of each subject photo. Yu et al. [54] proposed a model to synthesize sketches from photos by GAN that is assisted by composition information of the input photos. Their work removed blurs and spurious artifacts from the result sketches. Similarly, Lin et al. [55] presented a model to synthesize de-blurred sketches by deep CNN focusing on the estimation of motion blur. Zhu et al. [56] presented a model involving three GANs, in which each network gains knowledge of the photo–sketch pairs and imparts the learned characteristics to resulting sketches directly by a teacher GAN or by the comparison of the two student GANs. Radman et al. [57] proposed a sketch synthesis scheme based on the bidirectional long-short term memory (BL-STM) recurrent neural network.

## 3. Materials and Methods

The proposed framework comprises two neural nets. The first part is a compiler network C, which is based upon a residual network of two branches, and the skip connections are made in a spiral fashion. It is derived from [58], which was employed for neural-style transfer. For an input photo p, this part generates a raw sketch named s. The second part of the scheme is a feature-extractor called F, based on a pre-trained Vgg-19 network [59]. These net and associated components formulate another intermediate entity, called feature-sketch f. This composition is shown in Figure 1. The last step of the setup is a customized convolutional neural network, called discriminator **D**, to undertake a comparison between raw sketch s and feature sketch f. Their difference, combined with other loss functions, is then used to modify the weights of **C** and **D** networks iteratively during the training process. At end of the training, the C network is solely used to synthesize automated sketches from the test photos.

*Phase-1. Treatment of Images*: Photos/sketches of CUHK and AR datasets are already aligned, and they are of size 250 × 200 pixels. Therefore, they do not need any pre-processing. Photos and viewed sketches of XM2VTS and CUFSF datasets were not aligned. The following operations are executed upon the photos/sketches:Sixty-eight face landmarks on the image are detected by the dlib1* library.The image is rescaled in a manner that the two eyes are located at (75; 125) and (125; 125), respectively.The resulting image is cropped to a size of 250 × 200.

*Phase-2. Development of Feature Dictionary*: Patch matching is a time-consuming process. In addition, as already shown by the exemplar-based approaches, the computation of features for patches is resource intensive when conducted at run-time. Therefore, a dictionary of features of patches for all the images, including photos and viewed sketches in the reference set, is pre-computed and stored as a reference bank. Moreover, the entire length of reference sketches is not searched for a possible match. Instead, initially *n* top suitable candidate sketches to each input photo are selected at run-time based on their cosine distance at ReLU-5-1 features of the Vgg-19 net. Patch matching is then restricted within these *n* reference photos (*n* = 5 was used in all training runs of all iterations).

### 3.1. Compiler Network C

This network is composed of two identical strains, and each strain is composed of three stages. The first part consists of convolutional layers; it has residual blocks in the middle section and up-sampling layers in the end part. The structure is shown in Figure 2. It is a modified form of U-Net proposed by [58] for image style transfer and super-resolution. To introduce diversity and depth in the network, in a novel fashion, the skip connections in this model are added to an alternate strain instead of the original line. Therefore, each stage of the network on the left side is connected to the corresponding stage on the right side of the network and vice versa. The resulting shape looks similar to a spiral and, therefore, this construct is called Spiral-Net. Skip connections are added in this manner to (a) increase the width of each layer of the net, (b) augment feature matrices at different layers with new feature values from the other strain, and (c) populate feature matrices at different layers such that any half of the matrix vanishing due to ReLU and pooling operations may be repopulated with feature values. The last objective breaks any build-up of monotonous behavior due to ReLU and pooling operations. The compiler network C is a decisive module of this framework, and it plays major role during the implementation and operation phases. During the training phase, the training photo images are fed to this network and a pseudo sketch is formulated at its end. This sketch is further compared by the remaining parts of the overall scheme. Similarly, during the testing phase, a test photo is input to this network and its output is a synthesized sketch.

### 3.2. Feature Extractor F

A pre-trained model of Vgg-19 is used to extract features of the top *n* candidates of viewed sketches from the reference dataset for each train photo, where *n* can be set to any value, preferably between 5 and 10. Then input photos and the sketches are divided into identical maps/matrices of overlapping patches. An exemplar approach of the Markov random fields from [60] is preferred here, and it dictates that for each patch of the input photo, any of the candidate patches from the five sketches are selected based upon the shortest distance. This procedure is repeated from the first to the last patch of the input photo. Hence, F shapes up corresponding patches in a proper sequence to yield a feature map that is a representation of the intermediate sketch and is not exactly an image. It is used for comparison with the output of the compiler C through the discriminator **D**. The loss functions based on these comparisons are used to alternately update **C** and **D** networks.

Consider the given dataset as a universal set R composed of p photos and s sketches, where $R={\left\{\left(p_{i}^{R},\;s_{i}^{R}\right)\right\}}_{i=1}^{N}$. Here, N is the total number of photo–sketch pairs in the dataset. **F** aims at formulating a feature map $\theta^{l}\left(p\right)$ for the input photo p. $\theta^{l}\left(p\right)$ is used to augment the synthesis of the sketch $\hat{s}$. The MRF principle of [39] is applied to compose a local patch representation of p. It consists of the following stages:To begin with, p is input to the pre-trained Vgg-19 net.The feature map $\theta^{l}\left(p\right)$ is extracted at the l-th layer, where $l=\left(1,2,3,4,5\right)$, corresponding to (relu1−1, relu2−1, relu3−1, relu4−1, relu5−1) of F.A dictionary/look-up repository of reference representations is built for the entire dataset in the form of ${\left\{\theta^{l}\left(p_{i}^{R}\right)\right\}}_{i=1}^{N}$ and ${\left\{\theta^{l}\left(\;s_{i}^{R}\right)\right\}}_{i=1}^{N}$.Let us assume an r*r patch centered at point j of $\theta^{l}\left(p\right)$ as $T=\Omega_{j}\left(\theta^{l}\left(p\right)\right)$. Let us also assume corresponding patches $P=\Omega_{j}\left(\theta^{l}\left(p_{i}^{R}\right)\right)$ and $S=\Omega_{j}\left(\theta^{l}\left(s_{i}^{R}\right)\right)$ from the entire dataset.For every patch $T_{j}\;$, where $j=\left\{1,2,3,\ldots ,u\right\}$ and u is explained by the relation $u=\left(H^{l}-r\right)*\left(W^{l}-r\right)$, where $H^{l}$ and $W^{l}$ are the height and the width of the map $\theta^{l}\left(p\right)$, respectively, we find its closest patch $P_{j\prime }=\Omega_{j\prime }\left(\theta^{l}\left(p_{i\prime }^{R}\right)\right)$ from the look-up repository or dictionary based on the cosine distance.The cosine distance is defined with the help of Equation (1).
(1)$$\left(i\prime ,j\prime \right)=\frac{T_{j}{*P}_{j\prime }}{{\left|T_{j}\right|}_{2}*{\left|P_{j\prime }\right|}_{2}}$$
(2)$$\left(i\prime ,j\prime \right)=\underset{}{{\arg\max}_{j^{*}=1\sim m}^{i^{*}=1\sim N}}\frac{\left[\Omega_{j}\left(\theta^{l}\left(p\right)\right)\cdot \Omega_{j^{*}}\left(\theta^{l}\left(p_{i^{*}}^{R}\right)\right)\right]}{{||\Omega_{j}\left(\theta^{l}\left(p\right)\right)||}_{2}\cdot {||\Omega_{j^{*}},\left(\theta^{l}\left(p_{i^{*}}^{R}\right)\right)||}_{2}}$$Photos and sketches are aligned in the reference set. We index directly the corresponding feature patches $M_{j\prime }=\Omega_{j\prime }\left(\theta^{l}\left(s_{i\prime }^{R}\right)\right)$ for identified patches $P_{j\prime }=\Omega_{j}\left(\theta^{l}\left(p_{i\prime }^{R}\right)\right)$ by Equation (2).Successively, $M_{j\prime }=\Omega_{j\prime }\left(\theta^{l}\left(s_{i\prime }^{R}\right)\right)$ is used in place of every $T_{j}=\Omega_{j}\left(\theta^{l}\left(p_{j}\right)\right)$ to formulate a complete feature representation or the feature sketch at given layer l. Therefore, $F=\Omega_{j\prime }{\left\{\theta^{l}\left(s\right)\right\}}_{j=1}^{u}$.

### 3.3. Discriminator D

It is a basic convolutional network composed of six layers. Outputs of C and F networks are input to this net. This error, in addition to the other factors discussed later, is used to train the **C** network.

### 3.4. Loss Function

Feature Loss: The difference between the raw sketch s and the feature map f is expressed by a feature loss.
(3)$$F_{p}=\overset{5}{\underset{l=3}{\sum }}\overset{m}{\underset{j=1}{\sum }}{\left[\Omega_{j}\left(\theta^{l}\left(\hat{s}\right)\right)-\Omega_{j}^{\prime }\left(\theta^{l}\left(p\right)\right)\right]}_{2}^{2}$$
where l=3,4,5 refers to layers relu3-1, relu4-1, and relu5-1, respectively. High-level features after relu3 1 are better representations of textures and more robust against appearance changes and geometric transforms [60]. Features of the initial stages, such as relu1-1 and relu2-1, do not contribute to sketch textures well. Features extracted at a higher stage of the network, e.g., relu5-1, can better preserve textures. As a trade-off, r=3,4,5 is set to improve the performance of the setup and to decrease the computational overhead cost of patch matching procedures.

GAN Loss: The least-squares loss was employed when training the neural networks of the proposed setup. It is called LSGAN according to [61]. Equations (4) and (5) give the mathematical relationship of loss parameters/terms.
(4)EGAND=12ℚs~FsketchsDs−12+12ℚp~FphotopDGp2
(5)EGAN_G=12ℚp~FphotopDGp−12

Total Variation Loss: Sketches generated by a CNN network, used here as the discriminator **D,** may be noisy; and they may also contain unwanted artifacts. Therefore, according to previous studies [58,60,62], the total variation loss term was used. It was included to offset the possibility of noise and to improve the quality of the sketch. Its relationship is given by Equation (6).
(6)$$E_{\mathrm{tv}}\left(\hat{s}\right)=\underset{x,y}{\sum }\left({\left({\hat{s}}_{x+1,y}-{\hat{s}}_{x,y}\right)}^{2}+{\left({\hat{s}}_{x,y+1}-{\hat{s}}_{x,y}\right)}^{2}\right)$$

Here, ${\hat{s}}_{x,y}$ denotes the intensity value at $\left(x,y\right)$ of the synthesized sketch $\hat{s}$.
(7)$$E_{G}=\delta_{p}F_{p}+\delta_{\mathrm{adv}}E_{\mathrm{GAN}\_G}+\delta_{\mathrm{tv}}E_{\mathrm{tv}}$$
(8)$$E_{D}=E_{\mathrm{GAN}\_D}$$

## 4. Results

In this section, a detailed account of the implementation scheme is given. Moreover, it mentions the quality parameters used during this project and, finally, it elaborates upon the evaluation of the performance of the proposed and reference methods.

### 4.1. Datasets

Initially, two public datasets, namely CUFS and CUFSF [63], were employed. Then, the implementation was repeated with the augmentation of these two datasets by part of another set, called DIIT [64]. The details of repeated implementation are provided in Section 4.8 and onward. The composition and training–testing split of these datasets is given in Table 1. CUFSF is more challenging since its photos were captured under different lighting conditions and its viewed sketches show deformations in shape versus the original photos to mimic inherent properties of forensic sketches.

### 4.2. Performance Measures

This section describes those parameters that were selected to gauge the performance of existing and proposed methodologies.

Structure Similarity Index: The SSIM [67] gives a measure of visual similarity between two images. It is included here due to its prevalent use in state of the art, but we did not rely upon it as the decisive factor. The mathematical relationship of the SSIM is reproduced here, as Equation (9), from [67]. The value of the SSIM varies between −1 (for totally different inputs) and +1 (for completely identical inputs). Generally, an average value of SSIM scores for respective techniques over a specific dataset is computed to enable their direct comparison with each other.
(9)$$SSIM_{}\left(P,Q\right)=\frac{(2h_{p}h_{Q}+K_{1})\left(2Z_{PQ}+K_{2}\right)}{(h_{p}^{2}+h_{Q}^{2}+K_{1})(Z_{P}^{2}+Z_{Q}^{2}+K_{2}）}$$

Feature Similarity Index: The FSIM [68] is a measure of perceptual similarity between two images. It is based upon phase congruence and gradient computations and their comparison in respect of the given images. The FSIM is considered here as a reliable measure of similarity between synthesized sketches and their viewed sketch counterparts. The Feature Similarity Index (FSIM) [68] is a quality metric for two images based on their respective frequency dynamics, called phase congruency (PC), which is then scaled by the gradient magnitude (GM) of light variations of sharp edges at the feature boundaries. It is based on the premise that the human vision system (HVS) is more susceptible to frequency variations (PC) of low-level features in the given image. PC is, however, contrast invariant, whereas information of color or contrast affects the HVS perception of image quality. Therefore, the image gradient magnitude (GM) is employed as the second feature in the FSIM. Inherently, the FSIM is largely invariant to magnitude diversity.

PC and GM play complementary roles in characterizing the image’s local quality. PC is a dimensionless parameter defining a local structure. The GM is computed by any of the convolutional masks, such as Sobel, Prewitt, or any other gradient operator. The SSIM compares two images based on their luminance components only, while the FSIM considers the chromatic information in addition to the luminance of colored images.

The FSIM is computed by the following relations according to [66]: p(x) and q(x) are two images. PC_p_ and PC_q_ are their phase congruency maps, and Gp and Gq are their gradient magnitudes, respectively. Sim_PC_ is the similarity between these two images at point x, given by Equation (10) here. Sim_G,_ as mentioned in Equation (11), is their similarity based on the GM only, and Sim_L_ is their combined similarity at the point of consideration. Sim_L_ is measured by the relation given in Equation (12).
(10)$$Sim_{PC}\left(x\right)=\frac{2PC_{p}\left(x\right).PC_{q}\left(x\right)+C_{1}}{PC_{p}^{2}\left(x\right)+PC_{q}^{2}\left(x\right)+C_{1}}$$

C_1_ is a constant to ensure the stability of Equation (10).
(11)$$Sim_{G}\left(x\right)=\frac{2G_{p}\left(x\right).G_{q}\left(x\right)+C_{2}}{G_{p}^{2}\left(x\right)+G_{q}^{2}\left(x\right)+C_{2}}$$

C_2_ is a constant to ensure the stability of Equation (11).
(12)$$Sim_{L}\left(x\right)=[Sim_{PC}\left(x\right){]}^{\alpha }.[Sim_{G}\left(x\right){]}^{\beta }$$

The values of α and β are adjusted according to the importance of PC and GM contributions. Having determined the SimL at a given point x, the FSIM is computed for the overall domain of p(x) and q(x) images.
(13)$$FSIM=\frac{\sum_{x\in Ω}S_{L}\;\;\left(x\right)\;.\;Sim_{PCm}\;\left(x\right)\;}{\sum_{x\in Ω}\;Sim_{PCm}\;\left(x\right)}$$ where $\;Sim_{PCm}\;\left(x\right)=max\{PC_{p}\left(x\right).PC_{q}\left(x\right)\}$ is the maximum value in Equation (13).

### 4.3. Face Recognition

Face recognition is an important step in the existing state of the art to either determine or validate the efficacy of a proposed methodology of face sketch synthesis. Null-Space Linear Discriminant Analysis (NLDA) was employed to compute the quality of synthesized sketches for face recognition. Training and testing split of the total images to train and run the NLDA scheme is given in Table 2 and Table 3. Identical parameters were used during the application of the NLDA process to all sketch methodologies under test. In the repeated implementation OpenBR methodology [69] of face, recognition was additionally employed to ascertain the efficacy of proposed and existing schemes of face sketch synthesis.

### 4.4. Hardware and Software Setup

The compiler C and the discriminator D were updated alternately at every iteration. Neural networks were trained in two parts. In the first run of the setup, the CUFS reference style was used, and in its second part, the system was trained with the CUFSF reference style. In each case, however, the training photo–sketch pairs from both datasets were used. The different parameters and the associated information of training processes are given in Table 2.

### 4.5. Evaluation of Performance on Public Benchmarks

During the evaluation, we used photos from the CUFS dataset only to test the setup trained in the CUFS reference style. Similarly, photo–sketch pairs of the CUFSF dataset were used to test the proposed model trained in the CUFSF style. To determine the effectiveness of this model, results were compared with nine techniques of face sketch synthesis. They are MRF [39], MWF [40], SSD [17], LLE [38], FCN [44], GAN [45], RSLCR [42], Face2Sketch [25] (which contained a U-Net called SNET by its authors), and BiL-STM [57]. Synthesized sketches of the first seven techniques are available at [70]. We implemented the eighth method, Face2Sketch, ourselves in the PyCharm/UBUNTU environment assisted by NVIDIA GPU, mentioned in Table 2. The sketches were synthesized according to the training/testing parameters specified by its original work. Then SSIM, FSIM, and face recognition scores were computed by using these results of the eight techniques and reference sketches in MATLAB/Windows environment. Moreover, training and testing splits were fixed and identical for all the methods during computation of face recognition scores by the NLDA procedure. This detail is given in Table 4 and Table 5.

### 4.6. Results of CUFS Dataset

Table 6 shows that the SSIM values of SSD, Face2Skecth, RSLCR, and Spiral-Net are in the same range. Other methods scored less. The SSIM is a too generic a quality parameter to ascertain the visual similarity of images [47,71,72]. It was included in our work for comparison with the results of the previous works. Additionally, the feature similarity measure was computed for these sketch generation methods. Table 6 indicates that the FSIM metrics achieved by Face2Sketch and Spiral-Net are almost identical to each other and slightly higher than the other algorithms. Their difference from other methods’ FSIM score is 1–3% higher. In general, all these methods performed fairly similarly in terms of the CUFS dataset, where the viewed sketches lack any difference from the original photos and any variation in light intensity. Computations of the CUFS dataset were included to maintain a harmony of comparison with the previous works.

Table 7 records face recognition scores of these methodologies with help of the NLDA procedure, constituted of 142 features/dimensions of the images. Its graphical presentation is in Figure 3. RSLCR, Face2Sketch, and Spiral-Net performed superior to other methods. It is also evident that sketches synthesized by Face2Sketch and Spiral-Net contain more subtle information of the subject persons as compared to other methods since the former two algorithms attain 97% accuracy at 95 dimensions versus the 98% score of RSLCR at 142 dimensions. This improvement in the result also means lesser time complexity of the two methods to reach a rank-1 recognition level.

### 4.7. Results of CUFSF Dataset

SSIM, FSIM, and NLDA scores were computed for all eight methodologies. keeping reference parameters identical and intact for all. These values of BL-STM [34] were copied from the original paper. Table 8 records SSIM and FSIM scores of these algorithms for the CUFSF dataset. This dataset contains a diversity of age and ethnicity. Moreover, the viewed sketches were drawn with slight intentional deformations from the photos to render them similar to the properties of forensic sketches. It was observed that SSIM values did not convey any decisive information about the efficacy of the methodologies. RSLCR scored the highest in comparison to other algorithms. The FSIM was considered to be more robust a quality measure. Some of the exemplar-based methods, such as MRF, MWF, and LLE, achieved a 66% score, at par with the Face2Sketch method, which is based on a learning algorithm. The GAN method scored 67%, and it is also based on the neural network. It is seen that the proposed method of Spiral-Net achieved the highest value, of 68%, indicating that sketches synthesized by these methods contain more information of edges, contours, and shapes according to the original photo–sketch pairs.

The NLDA procedure was conducted using up to 300 features/dimensions as a validation step of face recognition in respect of all eight methods. Table 9 highlights those scores, and it is also shown graphically by Figure 4. Of the exemplar-based methods, MWF and RSLCR gained high scores, with 74.15% and 75.94% at 293 and 296 dimensions, respectively. Spiral-Net gained a competitive score of 73.14% at 44 dimensions, and it is equal to the Face2Sketch method, which scored equally at 217 dimensions. Therefore, Spiral-Net synthesizes sketches with enhanced features for a dataset that is considered challenging in the state of art. The best score of Spiral-Net is 78.4% at 184 features, and it further establishes the fact that the proposed method can imitate and “learn” subtle properties of the drawing style of the artist during this method’s training phase with photo-viewed–sketch pairs. It achieved 3–7% improvement over competitive methods from the exemplar-based domain (MWF, RSLCR) and the learning domain (GAN, Face2Sketch). It is seen that layers of the compiler C network from the first stage to the later stages were connected in a novel manner as alternate connections. This feature reduced the possibility of the development of monotony of values at subsequent stages since dissimilar layers were connected to each other progressively. As a result, the values in the matrices of layers bear significance, containing information of high-level features of the input photo or a sketch. This, in turn, preserves subtle information of each image throughout the progress of the network. Therefore, as a performance measure, sketches synthesized by Spiral-Net match better with the test photos at lesser dimensions by the NLDA scheme of face recognition as compared to sketches by other techniques.

### 4.8. Augmented Dataset and New Implementation

We introduced a new dataset from DIIT [64] and added its 234 photo–sketch pairs to the CUFS and CUFSF datasets. This exercise aimed to test our reference and modified schemes on hybrid datasets to verify their accuracy and to check their comparative performance. Detail is given in Table 8.

*Preprocessing of Augmented Datasets.**Phase-1. Treatment of Images.* Pre-processing steps of alignment and rescaling of the images were conducted according to Section 4.2, discussed above. *Phase-2. Development of Feature Dictionary.* The initial run was conducted for each scheme of SNET and Spiral-Net to compute feature files for both photo sets and their corresponding sketch sets at layers relu3-1, relu4-1, and relu5-1. The pre-computed files provided by [25] were not useful since they did not cover an additional part of the dataset introduced by this work. NOTE: The remaining parts of the implementation were conducted similar to Section 4.2, Section 4.3 and Section 4.4, as discussed above.

### 4.9. Evaluation of Augmented Datasets

The following text discusses the analysis of the results from experiments conducted on the augmented dataset.

▪It is important to note that we cannot compare newer results with any previous work since our modified or augmented dataset is put to use for the first time.▪The setup was implemented for two schemes, namely Face2Sketch (containing SNET as its component) and Spiral-Net. Therefore, the results may be compared between these two techniques.▪The second and third columns of Table 9 relate to these results. The second column gives values of the SNET technique, and the third column depicts result values for the Spiral-Net technique. It is seen that values of the SSIM and the FSIM for Spiral-Net are superior to those of SNET, which means that the proposed setup imparts more accuracy of features to the formulated sketches. Similarly, the face recognition values by NLDA and OpenBR methods for Spiral-Net are better than those for SNET by almost 2% and 5%, respectively. However, this improvement is achieved at the cost of processing time per photo since Spiral-Net contains almost double the layers of SNET (see Table 9).▪It is also observed from columns fourth and fifth, related to the VSF data component employed by SNET and Spiral-Net, respectively, that there is no marked difference of values between the two techniques. It indicates that CUFSF is inherently a challenging dataset since it copies the characteristics of real-life forensic sketches. Therefore, more research effort is required to fine-tune proposed and other new techniques to improve upon results of a singular CUFSF dataset or any combination of sets involving CUFSF.

## 5. Conclusions

In this work, a novel architecture of U-Net comprising two strains instead of one for the forward pass was proposed. Moreover, the skip connections were made cross-wise between the two strains to reduce the possibility of any monotonous build-up of feature values due to ReLU and pooling operations. Experimental results in comparison to exemplar-based and learning-based schemes indicated that the proposed setup enhances the performance benchmark of sketch synthesis by around 5%. Moreover, a newer approach of augmented datasets comprising conventional sets from CUFS/CUFSF and a part of the DIIT photo–sketch dataset was also applied. Then, it was demonstrated that our modified Spiral-Net achieves a superior performance by 5% compared to its original framework of U-Net. In the future, the authors plan to conduct further experimentation to improve the discriminator D neural network of this framework so as to further refine the loss functions of the technique. Moreover, the currently used feature extractor may be replaced with the neural architecture proposed by Li et al. [73,74].

## Figures and Tables

**Figure 1 sensors-21-08178-f001:**
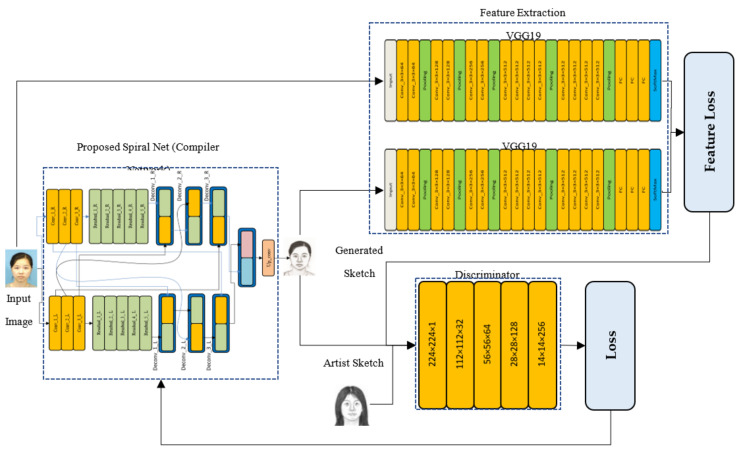
Schematic diagram of the proposed method.

**Figure 2 sensors-21-08178-f002:**
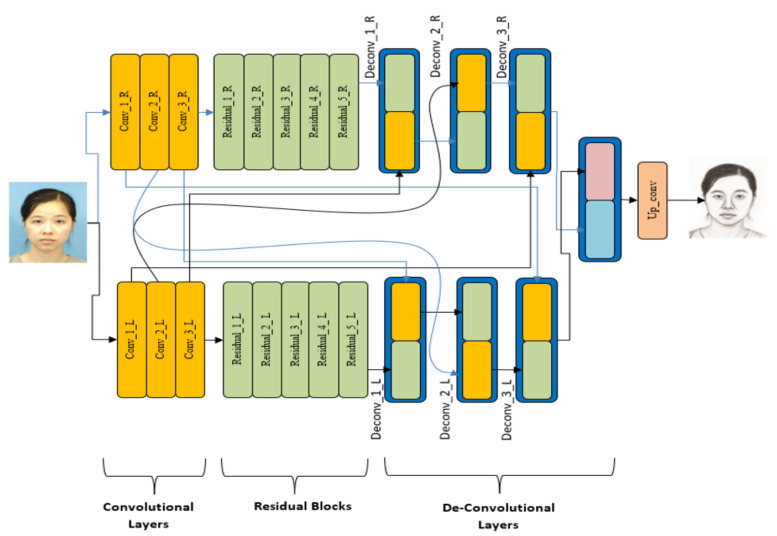
Architecture of Spiral-Net as a sketch compilation network.

**Figure 3 sensors-21-08178-f003:**
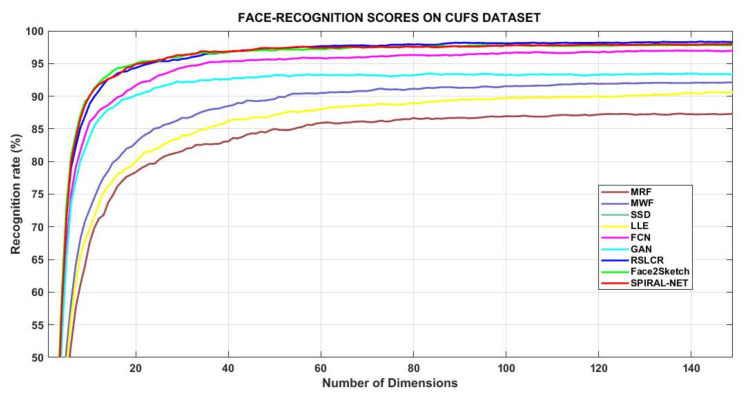
Comparative view of NLDA scores by different techniques on CUFS dataset.

**Figure 4 sensors-21-08178-f004:**
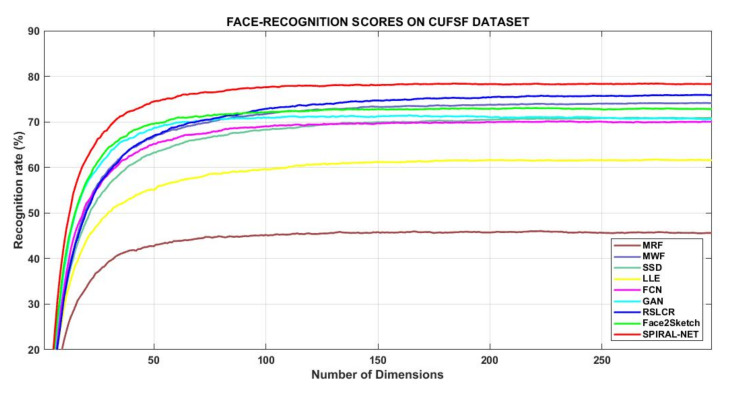
Comparative view of NLDA scores by different techniques on CUFSF dataset.

**Table 1 sensors-21-08178-t001:** Details of initial datasets.

Dataset		Total Pairs	Train	Test
CUFS	CUHK [37]	188	88	100
	AR [65]	123	80	43
	XM2VTS [66]	295	100	195
CUFSF		1194	250	944
Total Pairs		1800	518	1282

**Table 2 sensors-21-08178-t002:** Parameters for processing.

S No	Item		CUFS	CUFSF
1	Hardware		Core i-7 ^®^, 7th Gen, NVIDIA 1060 (6GB) GPU	
2	OS		Ubuntu Linux	
3	Environment		PyCharm (CE), Torch 1.4.0	
4	Moderating Weights	$\delta_{p\prime }$	1	1
		$\delta_{\mathrm{adv}}$	10^3^	10^3^
		$\delta_{\mathrm{tv}}$	10^−5^	10^−2^
5	Learning Weights		10^−3^ to 10^−5^ reducing by a factor of 10^−1^	
6	Batch Sizes		4 to 2 for different iterations	
7	Processing Time		See respective tables	

**Table 3 sensors-21-08178-t003:** Distribution of synthesized sketches by the NLDA procedure of face recognition.

Dataset	Total Pairs	Train	Test
CUFS	338	150	188
CUFSF	944	300	644

**Table 4 sensors-21-08178-t004:** Comparison of SSIM and FSIM Values for CUFS.

Type	MRF [10]	MWF [11]	LLE [9]	SSD [4]	FCN [15]	GAN [16]	RSLCR [13]	Face2Sketch [6]	BiL-STM [28]	Proposed Spiral-Net
Proc Time (msec/photo)	Not presented by the original works									7.57
SSIM	51.31	53.92	52.58	54.19	52.13	49.38	55.71	54.41	**55.19**	54.42
FSIM	70.46	71.45	70.32	69.59	69.36	71.54	69.66	**72.59**	67.77	**72.50**

**Table 5 sensors-21-08178-t005:** Comparison of face recognition scores for CUFS.

Type	MRF [10]	MWF [11]	LLE [9]	SSD [4]	FCN [15]	GAN [16]	RSLCR [13]	Face2Sketch [6]	BiL-STM [28]	Proposed Spiral-Net
NLDA Score (Equal/Best)	87.34	92.10	90.61	90.61	96.99	93.48	**98.38**	97.82	94.87	**97.04**/97.23
No. of Features (Equal/Best)	138	148	144	144	137	139	142	95	-	**95**/148

**Table 6 sensors-21-08178-t006:** Comparison of SSIM and FSIM Values for CUFSF.

Type	MRF [10]	MWF [11]	LLE [9]	SSD [4]	FCN [15]	GAN [16]	RSLCR [13]	Face2Sketch [6]	BiL-STM [28]	Proposed Spiral-Net
Proc Time (msec/photo)	Not presented by the original works							4.37	-	7.89
SSIM	35.36	40.83	39.66	41.88	34.39	34.81	42.69	38.97	**44.56**	38.32
FSIM	66.06	66.76	66.89	64.81	62.91	67.05	63.16	66.87	68.04	**68.10**

**Table 7 sensors-21-08178-t007:** Comparison of Face Recognition Scores for CUFSF.

Type	MRF [10]	MWF [11]	LLE [9]	SSD [4]	FCN [15]	GAN [16]	RSLCR [13]	Face2Sketch [6]	BiL-STM [28]	Proposed Spiral-Net
NLDA Score (Equal/Best)	46.03	74.15	70.92	61.76	70.14	71.48	73.05/75.94	73.05	71.35	73.14/**78.42**
No. of Features (Equal/Best)	223	293	266	274	226	164	102/296	217	-	**44**/**184**

**Table 8 sensors-21-08178-t008:** Details of augmented datasets.

Dataset		Total Pairs	Train	Test
VSC	CUHK [37]	188	88	100
	AR [65]	123	80	43
	XM2VTS [66]	295	100	195
	IIIT-D	234	94	140
Total Pairs		840	362	478
VSF	CUFSF	1194	250	944
	IIIT-D	234	94	140
Total Pairs		1428	344	1084

**Table 9 sensors-21-08178-t009:** Comparative values of performance for augmented datasets using SNET and proposed Spiral-Net.

Type	VSC-SNET	VSC-Spiral-Net	VSF-SNET	VSF-Spiral-Net
Proc Time (msec/photo)	4.3033	8.5619	4.3113	8.1858
SSIM	38.18	**46.81**	40.33	40.51
FSIM	67.65	**68.34**	70.25	70.13
NLDA Score (1998) (%)	67.82	**69.61**	65.99	65.44
OpenBR_FR Score (2013) (%)	66	**71.3**	30.7	30.4

## Data Availability

Not applicable.

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
