# Peer review of "A Decision Support System for Face Sketch Synthesis Using Deep Learning and Artificial Intelligence"

_sensors, 2021, doi:10.3390/s21248178_

Round 1

Reviewer 1 Report

The authors proposed a new-found architecture of the neural network, which is called Spiral-Net that is a modified version of U-Net to perform face sketch synthesis. Whether in innovation or in some language presentation, it needs further improvement.

  1. Don't abbreviate some professional words for the first time, such as IoT and NLDA.....
  2. Ablation study is needed in this paper.What is the difference between the newly created network structure and the basic framework? Why are there performance differences? What module plays a decisive role? Can the module can be further verified by ablation study?
  3. The conclusion section is needed to further improvement. Future research plans should be introduced.
  4. Some recent research advances need to be discussed. For example: Self-Selection Salient Region-Based Scene Recognition Using Slight-Weight Convolutional Neural Network; Improved performance of face recognition using CNN with constrained triplet loss layer; An end-to-end trainable multi-column cnn for scene recognition in extremely changing environment .........

Author Response

Response sheet has been added

Reviewer 2 Report

In this paper, the authors propose an approach for face sketch synthesis using deep learning. 

The topic proposed by the authors is extremely timely and the approach they propose is interesting. 

The authors describe their approach providing all the technical details. This contributes to make the paper complete and the approach convincing. On the other hand, however, it makes it difficult for the reader to follow all the technical steps and the many formulas in the paper. For this reason, I suggest the authors to include a leading example. Such an example should be broken up throughout the paper to intuitively explain the various technical steps described.

The experiments appear convincing.

As a further request, since the authors insist a lot on the IoT context in which their paper is set, I suggest them to indicate how their approach can be used with newer and more innovative IoT architectures. In particular, I suggest them to consider Social IoT (described in the papers by Iera, Atzori and Morabito) and MIoT (Multi-IoT, not Medical IoT!!!) described in the papers by Baldassarre, Cauteruccio, Virgili and Fortino. This could be done by adding an appropriate paragraph in the related works, or in the conclusions, or even by adding a Discussion section.

Author Response

Response sheet has been added. 

Round 2

Reviewer 1 Report

The authors addressed all the problems I concerned. So, I suggest to accept this paper in the current form.

Author Response

Response sheet has been attached. thank you

Reviewer 2 Report

The authors have made a very little effort to comply with my suggestions. For instance, I suggested them to talk about the possibility to applu their approach in more advanced IoT architectures, like SIoT and MIoT (Multi-IoT). However, they performed no modification in this direction. Furthermore, I suggested the work of several authors (clearly different from myself) to consider, and to help them, I specified the name of these authros. However, they did not consider any of these authors.

Therefore, I vote again for a major revision of the paper but, the next time, if the authors do not consider my suggestions seriously I will vote for reject.

Author Response

(The authors gave the same response as above.)

Round 3

Reviewer 2 Report

The authors have performed the modifications I required in my previous review. So I'm satisfied for the new version of this paper.